# Emotional and Mental Nuances and Technological Approaches: Optimising Fact-Check Dissemination through Cognitive Reinforcement Technique †

**Francisco S. Marcondes** [1,*], **Maria Araújo Barbosa** [1], **Adelino de C. O. S. Gala** [2], **José João Almeida** [1] **and Paulo Novais** [1]

1 ALGORITMI/LASI, University of Minho, 4710-057 Braga, Portugal; pg42844@alunos.uminho.pt (M.A.B.); jj@di.uminho.pt (J.J.A.); pjon@di.uminho.pt (P.N.)
2 DIGIMEDIA, University of Aveiro, 3810-193 Aveiro, Portugal; adelinogala@gmail.com
* Correspondence: francisco.marcondes@algoritmi.uminho.pt
† This paper is an extended version of our paper published in Barbosa, M.A.; Marcondes, F.S.; Novais, P. Cognitive Reinforcement for Enhanced Post Construction Aiming Fact-Check Spread. In Proceedings of the International Symposium on Distributed Computing and Artificial Intelligence, Guimaraes, Portugal, 12–14 July 2023. Springer: Berlin/Heidelberg, Germany, 2023; pp. 203–211.

**Abstract:** The issue of the dissemination of fake news has been widely addressed in the literature, but the issue of the dissemination of fact checks to debunk fake news has not received sufficient attention. Fake news is tailored to reach a wide audience, a concern that, as this paper shows, does not seem to be present in fact checking. As a result, fact checking, no matter how good it is, fails in its goal of debunking fake news for the general public. This paper addresses this problem with the aim of increasing the effectiveness of the fact checking of online social media posts through the use of cognitive tools, yet grounded in ethical principles. The paper consists of a profile of the prevalence of fact checking in online social media (both from the literature and from field data) and an assessment of the extent to which engagement can be increased by using simple cognitive enhancements in the text of the post. The focus is on Snopes and 𝕏 (formerly Twitter).

**Keywords:** social networks; post generator; fact checking; emotions; personality; engagement





## 1. Introduction

The digital information age has created an overwhelming flow of data, making it harder than ever to separate fact from fiction. Social media platforms, particularly Twitter and Facebook, have become epicentres for the rapid dissemination of information, often without proper validation. Despite exhaustive research on these platforms, there remains a discernible bias towards them, inadvertently side lining other important platforms such as Weibo and VK, which are popular mainly in Eastern regions. This oversight calls for a comprehensive understanding of the nature and dynamics of data on these platforms. Moreover, the complex terrain of fact checking in today's digital space requires intricate navigation, weaving through technology, emotion and public perception. These multidimensional aspects of information dissemination, if not addressed holistically, could inadvertently lead to misleading narratives or perpetuate existing biases.

For reference, Table 1 shows a comparison between the sizes of the selected fact-checking accounts. Considering that the hype about fake news is still present, the growth rate of these accounts is worse than expected. The profile that has performed better is @APFactCheck. It is interesting to note that the engagement rate of @APFactCheck is significantly higher than that of the other profiles, but the number of followers is also significantly lower. However, it is the only account that has grown significantly over the period. So Table 1 might suggest that there is also a relationship between tone and number

of followers. Unfortunately, the data are not available, but if this assumption is correct, it could explain the reduction in appealing phrases in current posts compared to 2020. Finding a balance in tone seems to be a feature to consider when distributing fact checks. This may be related to the fact that people use Twitter in their rest time; then, catchy posts would often be preferred. For an illustration of two different tones, refer to Figure 1.

**Table 1.** Twitter account sizes of selected fact-checking agencies.

| Twitter Handle | Followers | | Tone | |
|---|---|---|---|---|
| | 2020 | 2023 | 2020 | 2023 |
| @Snopes | 237 K | 287 K | journalistic with witty phrases | journalistic with witty phrases |
| @PolitiFact | 673 K | 646 K | journalistic | journalistic with social interaction |
| @Poynter | 214 K | 216 K | journalistic | journalistic |
| @factcheckdotorg | 190 K | 190 K | journalistic | journalistic with witty phrases |
| @APFactCheck | 31 K | 78 K | tabloid | tabloid (less appealing) |
| @MBFC_News | 4 K | 5 K | only link | only link |
| @TheDispatchFC | 1 K | 2 K | only link | only link |

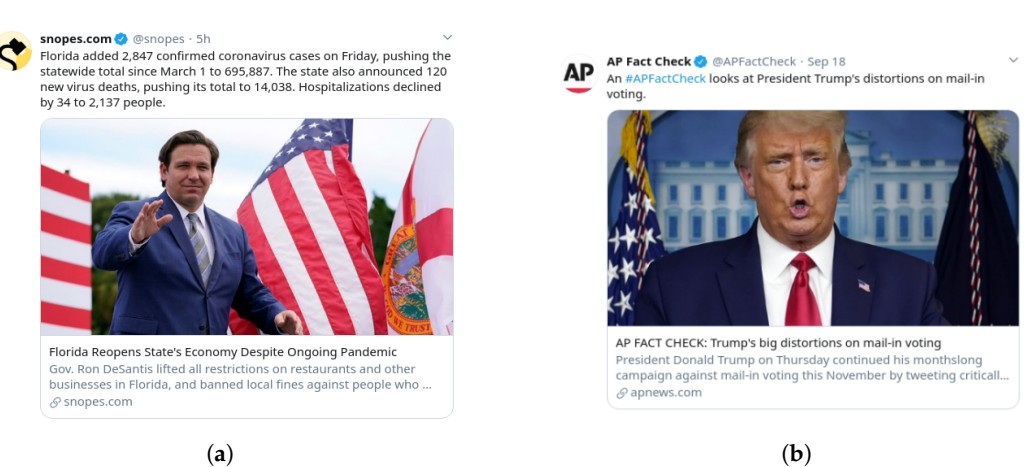

(**a**)            (**b**)

**Figure 1.** Examples of slightly different strategies for creating tweet text (both tweets are from 2020). (**a**) @snopes traditional journalism. (**b**) @APFactCheck provocative content.

By analysing the tone concept, it is possible to extract some features that can be used to tune a tweet to be more engaging. A common way to automate a Twitter account for posting news is to find a sentence in the body of the news that has the higher number of common words in the text (extractive summarisation [1]) and post it along with the link. As an illustration, Figure 2 shows two extractive summarisations of the same fact check, along with some of the heuristics discussed in [2]. The difference is that the first searched for common words in the text, whereas the second searched for common words associated with the highest surprise emotion.

This example suggests that it is possible to adjust the tone of a tweet to make it more engaging. The caveat, however, is not to fall into the tabloid trap, which might discourage people from becoming followers.

The objective of this paper is to explore the use of cognitive reinforcement to improve the dissemination of fact checks in online social media and thus underline the significance of the cognitive nuances to ensure that authentic information not only reaches its audience but also deeply resonates with them.

This paper is divided into three main sections. The first, Section 2, presents a literature review to present the current state of the art in fact-checking dissemination. The second, Section 3, builds on the previous one and presents a strategy for reinforcing tweets to increase the engagement rate. Finally, Section 4 presents a comprehensive view of fact-check propagation compared to the results obtained in the previous section.

We emphasise that this paper is an extended version of [3] (mostly covered in Section 3). To reduce the scope, the experiments are based on Twitter with fact checks from Snopes.

Also, Twitter has recently been renamed to $\mathbb{X}$, but since Twitter is still widely known, the old name is kept in this paper.

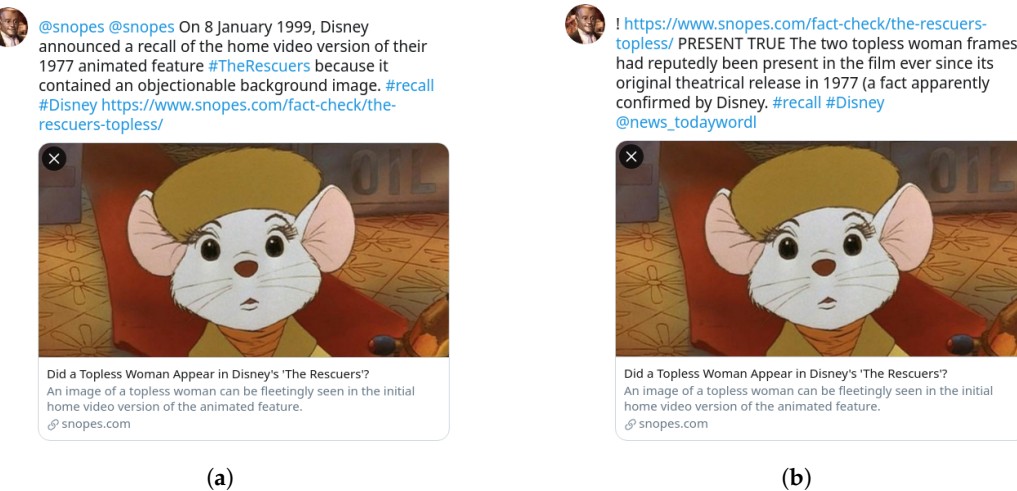

(**a**)  (**b**)

**Figure 2.** Illustration of the impact of emotion in a tweet tone. (**a**) Word-based approach. (**b**) Surprise maximisation.

## 2. Literature Survey

The first step in any research endeavour is to raise the relevant literature, searching for the state of the art in the field. Thus, this section presents a brief survey conforming to Kitchenham's guidelines [4], aiming to understand the relevant features for crafting an engaging fact-check post.

The period studied was between 2016 and 2021, using Scopus and ACM-DL. Note that 2016 is due to the quality shift that fake news underwent at that time, with the rise of Cambridge Analytica and the troll factory concept [5].

The query looked for fact checking, debunking and counter-propaganda spread, reach, and engagement on social media. To match the target, papers that aimed to identify, detect or assess fake news were filtered out. For the same reason, issues related to youth exposure, health, and democracy were also filtered out. Finally, COVID-19, which has been a major source of misinformation in recent years, was excluded from this study because it is expected to be a rare event.

During the screening process, it was found that keywords related to digital literacy and product reviews could also be excluded from the query. The keyword 'fake-news' was considered for exclusion, as it returned several false positives, but, as it also excluded papers of interest, it was decided to filter these papers during the screening process. Previously known papers that are also relevant but were not retrieved by the systematic search are also included in the sample. For some of these, forward snowballing was applied to retrieve additional related papers. The filtering procedure is detailed in Figure 3.

This survey is biased in the sense that only Twitter and Facebook data were retrieved and other social media, which are widely used to spread fake news, were not included. It also does not include information from Eastern social networks such as Weibo and VK.

After the search, it was realised that it is difficult to conduct a systematic review on fact checking due to the high number of false positives related to the detection or spread of fake news. Therefore, it took some time to realise that the number of papers discussing fact-checking dissemination strategies is small. The main findings are organised in a taxonomy and presented in Figure 4. In addition, it was also possible to perform a comparison between fact-check and fake-news features, which is presented in Figure 5. Overall, the combination of content characteristics, human cognitive biases, and the dynamics of social media contribute to the viral spread of fake news.

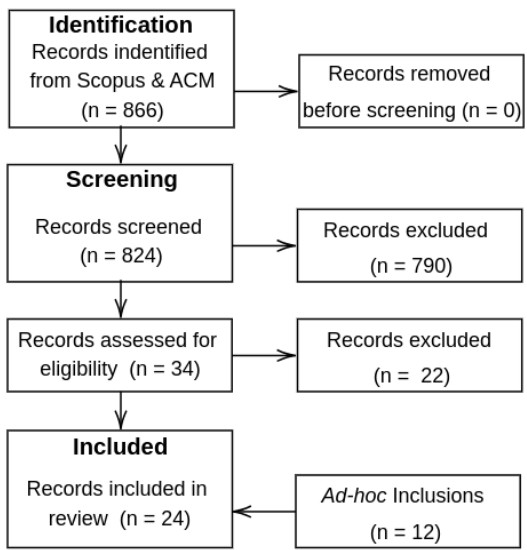

**Figure 3.** Filtering process and results.

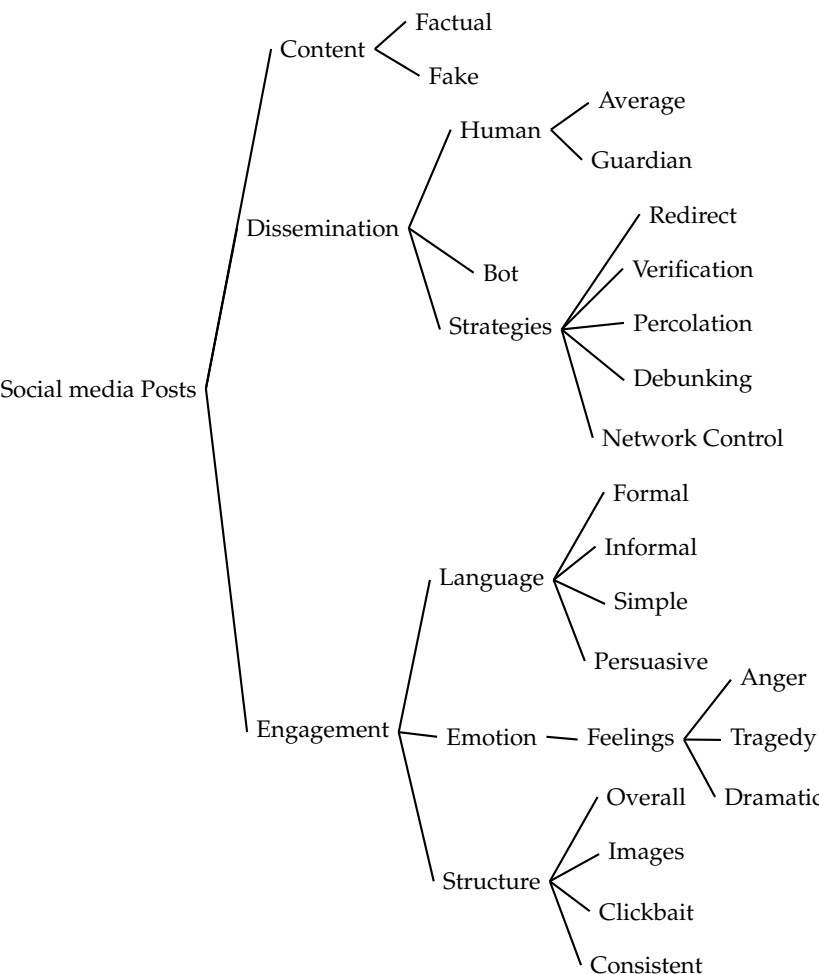

**Figure 4.** A taxonomy of the dissemination of facts in online social media extended from [6]. For factual, refer to [7–11]; fake, to [10,12–14]; average, to [15]; guardian, to [12]; bot, to [2,11,15]; redirect, to [16]; verification, to [17]; percolation, to [18]; debunking, to [19,20]; network control, to [21]; formal, to [9]; informal, to [13]; simple, to [13]; persuasive, to [13]; feelings, to [10,12]; overall, to [22]; images, to [14]; clickbait, to [13]; and consistent, to [11].

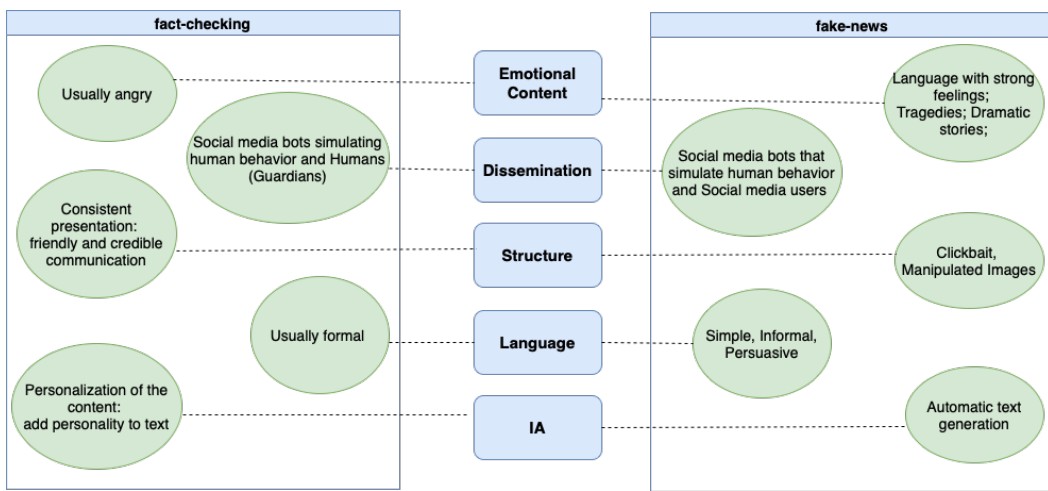

**Figure 5.** Comparison between fact-checking and fake news engagement properties [6].

The survey reveals the involvement of two distinct types of actors in the dissemination of fact-checked news: human and non-human. Human actors, exemplified by the Guardians, utilize social media profiles to share fact-checked article URLs [7]. However, even well-intentioned Guardians face challenges when countering highly sophisticated and technologically advanced propaganda structures [23]. This has led to the utilisation of non-human applications, such as social media bots, to enhance the dissemination of fact-checked posts on Twitter. These bots are algorithmically designed to mimic human behaviour, enabling them to generate content and engage with users on social media platforms. Given their substantial presence on Twitter, constituting approximately 9% to 15% of Twitter accounts [24], they are well positioned to expedite the spread of fact checking through actions like sharing and replying.

One effective property is to employ techniques that simulate human behaviour in order to foster user engagement with the post's content. This can be achieved by instructing the bot to perform actions commonly carried out by social media users to attract attention to their posts. For instance, identifying influential users through replies or mentions can encourage their involvement in the outreach process. Additionally, leveraging trending hashtags can also be beneficial. To establish a strong connection with users, it is important for bots to maintain consistency in their presentation, exhibit friendliness, and demonstrate credibility [11]. Another successful approach involves utilising bots to actively seek out posts containing false content and respond to them with accurate information, thereby enlightening users with the truth [11]. Although this property is commonly employed by fact-checking agencies, caution must be exercised when using negating words, as they can have an unintended effect on readers, especially if they are encountering such information for the first time.

In order to comprehend how a story becomes viral, it is essential to first establish the concept of spread. In the realm of social networks, the spread of a topic is often defined as a form of social contagion. Each person within a social network can be considered a node, connected to other nodes through social ties. When an individual encounters a piece of information, it can easily be transmitted to others who are socially connected [25]. This propagation continues in a cascade, intensifying as it reaches more interactions [26]. It is important to note that social media algorithms do not display every post simultaneously in users' feeds, and interactions play a crucial role in determining how many people will see a post. However, the initial user who comes across a post lacks the parameters, such as the number of interactions, to initiate a cycle that could lead to virality. Therefore, an alternative approach is to focus on the content itself and how the narrative is constructed. In this regard, ref. [12] discovered that, compared to typical responses on Twitter, fact-checking posts more frequently employ impersonal pronouns, adopt a more formal language style, demonstrate a clear intent to verify facts, and commonly employ adjectives such as fake, wrong, dumb,

false, and untrue. However, it is important to recognize that these characteristics are contrary to the expectations of social media users. For instance, the use of formal language may create a sense of detachment between the user and the posts, as it is not the type of content that typically engages social media users.

Another challenge that can arise is the mechanisation of text, which can create a disconnect between the content of a post and the reader [27]. This disconnect can make the content appear artificial and enter what is known as the "uncanny valley" [28]. To address this issue, two approaches have been proposed: incorporating personality into microblog text using the Big Five personality theory [27], and identifying the emotions expressed in tweets and their relationship to the potential for virality. Both approaches aim to capture the user's attention and make the text feel more human-like. Although these properties have not yet been specifically tested in the context of fact checking, some studies have explored the connection between the presence of certain emotions or personality traits and user engagement. Regarding emotions, tweets conveying melancholy are more likely to be retweeted and liked by users, while tweets expressing other emotions (such as anger) are less likely to receive engagement. On the other hand, tweets associated with corrections (e.g., tweets from fact-checking agencies) often contain expressions of anger [10]. This could potentially explain the limited interactions that fact-checking posts typically receive.

Several studies have focused on tackling the spread of misinformation and fake news on social media platforms, using different strategies and methodologies. One notable initiative is the Facebook Redirect programme, launched in partnership with Moonshot in 2020, which aims to steer users away from engaging with violent extremist content [16]. The programme uses a list of keywords related to dangerous individuals and organisations, a safety module with explanatory text and calls to action, and landing pages from delivery partners to provide support services. The success of this initiative was measured by the conversion of passive searches into active conversations for support and the positive engagement of tens of thousands of users.

Twitter has also been subject to research in combating misinformation. One study investigated the veracity of tweets related to the death hoax of Singapore's first Prime Minister Lee Kuan Yew [22]. Analysing 1000 tweets through logistic regression, the study found that tweet accuracy could be predicted based on factors like clarity, credible sources, and author attributes, like length of membership and followers. However, challenges such as username squatting, see [29], may complicate the interpretation of these attributes. Another approach involved studying fact-checking behaviour on Twitter, revealing that the simple tweet dissemination of fact checks might not effectively counteract misinformation [2]. An algorithmic model was proposed to simulate human-like behaviour, potentially enhancing the impact of counter-propaganda efforts.

Research has also investigated the dynamics of misinformation on platforms like Facebook [17]. A comparison was drawn between Facebook's efforts to combat misinformation and the Right-Click Authentication method, which empowers users to classify content based on existing tools. Another study proposed using bootstrap percolation to model fake news spread and fact-checking effectiveness [18]. The results demonstrated that addressing all sources of fake news may not be sufficient to reverse popular beliefs. Additionally, a model was developed to understand the spread of rumours in online social networks, emphasising the role of individual attitudes, technology use, and crisis management in controlling rumour propagation [19].

To counter rumours effectively, the concept of official rumour refuting information (ORI) was introduced, emphasising the importance of government credibility and the timely release of information [20]. Another research effort focused on using graph convolutional networks to detect rumours based on propagation patterns [21]. This method demonstrated improved accuracy compared to existing models in identifying different types of rumours by learning from the propagation structures within social networks.

As a result, the narrative presented by Jordan Ellenberg [30] regarding Abraham Wald's World War II observations aptly mirrors the current challenges in fact-check spread-

ing. The prevailing focus remains on countering widely disseminated disinformation, neglecting the underlying emotional and cognitive drivers that fuel its spread. Addressing these is imperative, as mere identification, regulation, and suppression might inadvertently bolster counterproductive disengagement sentiments.

Nguyen Vo and Kyumin Lee's [12] study sheds light on the potential pitfalls in current fact-checking methodologies on Twitter. The formal language and specific linguistic markers of these fact-check posts might inadvertently alienate users, thereby defeating their intended purpose. Existing research highlights the risks of mechanising text, which can alienate readers by coming off as artificial or uncanny [27,28]. Solutions include infusing textual content with identifiable personality traits or aligning them with emotional cues that resonate with users, enhancing engagement.

Interestingly, while certain emotions, such as melancholy and surprise, boost tweet engagement, others like anger have the opposite effect. This could elucidate the observed lower engagement levels with fact-checking posts known to convey anger [10]. Additionally, the epistemological scrutiny of fact checking [31] offers a nuanced perspective, highlighting the multi-dimensional nature of public discourse. This study's methodology aspires to bolster fact-checking dissemination, recognising the intricate tapestry of politics, psychology, information media, and technology that shapes public debates.

The viral spread of fake news is driven by a combination of factors that can be categorised into two main areas. The first area relates to the semiotic structure and content of fake news, including elements such as graphics, headlines and language that attract attention and encourage sharing on social media [9,13,32]. The second area relates to cognitive biases that make individuals susceptible to misinformation and lead them to share fake news without critically evaluating its accuracy [33,34]. Human tendencies towards emotional engagement, preference for sensational content and reliance on authoritative figures contribute to the success of fake news. This phenomenon is amplified by social media signals that reinforce perceptions of popularity, even when the information is false. Although both social bots and human users contribute to the spread of fake news, studies show that real people are more likely to retweet fake news than accurate stories.

In addition to the spread of fake news, social media has been used as a research environment for other computer science studies that require data collection or human testing. Developing research on social media can raise a number of complicated ethical issues related to data, consent, traceability or even the morality of individuals [14]. It seems that these issues are being recognised and that attempts are being made to develop strategies to mitigate them. One example is the platform described in [35], which simulates Twitter and allows social media studies to be conducted without compromising moral principles.

Fact-checking competes with misinformation in two dimensions: fact-checkers and fact-checking consumers [15]. The former are working hard to debunk a claim, but their efforts are struggling to keep up with the vast amount of misinformation. The fact-checking consumers share tweets that debunk the misinformation. In Addiction, the authors examine the strategies used by social bots to increase the spread of fake news and present three case studies: producing a large number of original tweets, using trending hashtags and injecting fake news content into conversations.

In sum, this search underscores the multifaceted nature of fact checking in the digital age, seeking to navigate the intertwined realms of technology, emotion, and public perception to foster a more informed global discourse.

It is worth mentioning that the mechanisms of different social media platforms that trigger their existence can be described by "the type of connection between users, how the information is shared and how users interact with the media streams" [36] as follows:

**Interest-graph media** Twitter, Gettr and similar platforms depend on users forming connections with others. One of the driving forces is shared interests, either formal or informal, in current situations. Connections do not always have to be reciprocated. The information shared is streamed through messages in reverse chronological order.

**Social networking sites (SNSs)** Facebook and similar platforms depend on users connecting with each other. People share information, comment on and respond to each other's posts. Short and informal narratives about users' lives and thoughts are shared. Status updates are sorted into a time-ordered stream for each user to read.

**Professional Networking Services (PNSs)** LinkedIn and similar platforms aim to provide services in the context of work. They depend on the user connecting to them. Typically, professional information, news and recommendations are shared. There is also a news stream that works similarly to SNSs.

**Content sharing and discussion services** YouTube, Vimeo and similar; SlideShare and similar; and discussion/review forums such as Reddit are also relevant. Users produce, share and discuss content by following channels, blogs or discussions that suit their preferences. Readers can also comment on these posts. Some sites such as YouTube create a time stream of videos.

The mediation algorithm is a tool used by social networks to build the user's feed (or timeline). It focuses on personalisation, with the aim of creating a feed that is well suited to each user, based on their previous activities. In other words, the feed should reflect the preferences and interests of each user [37].

At least in principle, mediation algorithms have no editorial line, despite blindly maximising user preferences. Yet, somehow, it favours toxic content and fake news [38]. The explanation is simple: the likelihood of responding to appealing content is high; responding tells the mediation algorithm that the topic is of interest. As a result, the feed will receive more and more content related to that topic. This is certainly naive behaviour. Understanding that engaging content increases engagement, Facebook, for example, increases the distribution of posts tagged with the angry emoji [39].

However, this is not the only variable to consider. Other variables that are likely to be highly weighted include novelty (avoid showing repeated content in the feed), recency (new content is often preferred) and avoidance of repetition (avoid repeating posts that have already been shown). These variables, especially for heavy users (users who spend a lot of time connected to the platform), require a large number of new posts to work.

New users of digital social networks perform searches using the algorithm with low expertise. This means that the scope of the search is defined using more general terms about the topic of interest. The result tends to be the most popular collection of news or disinformation about the topic. In a scenario where the amount of disinformation on a given topic is high and the user is not an expert on the topic, the spread of disinformation is therefore amplified, with the resulting cognitive effects.

### 3. Cognitive Reinforcement

Research indicates that distressing news and political content with strong emotional appeal are particularly effective in driving engagement with fake news [8,40]. Manipulative strategies, such as using attention-grabbing images and clickbait techniques, enhance the dissemination of false information. Furthermore, readers often react based on headlines and tags rather than thoroughly consuming the entire piece of content.

In other words, emotional drivers play a crucial role in the spread of misinformation and fake news, as well as in the effectiveness of countermeasures. A study focusing on the effectiveness of fact checking on social media [41] shows that such interventions have minimal impact on stopping the spread of misinformation. In particular, the fear of social isolation emerged as a significant deterrent, suggesting that social consequences measures, such as account suspensions, might be more effective. However, this first solution is a type of approach that could potentially inhibit free speech. The suggestions of the first article are in line with the findings of another study [42], which highlights the significant role of emotions, especially in believing fake news.

The study's findings show that a heightened emotional state, especially when relying on emotion over reason, correlates with increased susceptibility to fake news. Similarly,

research on the emotional framing of news [43] suggests that news stories that evoke approach emotions (anger and hope) engage readers more intensely than those that evoke avoidance emotions (fear) and influence their subsequent information-seeking behaviour. Finally, a study conducted in South Korea [44] found that anger specifically contributes to the spread of COVID-19 misinformation, with angry individuals being more likely to perceive false claims as credible, particularly among conservative audiences. Taken together, these findings underscore the importance of emotions in the spread and reception of fake news. Thus, any effective text reinforcement strategy aimed at mitigating the spread of fake news must take these emotional drivers into account and use them to increase engagement, critical thinking, and discernment among the public. The same can be inferred for other cognitive dimensions.

A tweet is an online publication that typically consists of four different elements: text-core, emoticons, hashtags and links. The text-core contains the textual part of the post, comprising the relevant content/message. An emoji is a pictogram that can be embedded in the text-core. Hashtags are search terms that can also be included in the post, and links are URLs to websites.

The source of the information is the Snopes fact-check reports. The information is extracted using a `beautifulsoup` scrapper, and the fields searched for are URL, date, newspaper name, title, claim, classification and the content of the report. The result is stored in a JSON file.

*3.1. Summarisation*

The aim of this section is to obtain the post element text-core. To guide this procedure, the following requirements are established:

- Be consistent. Avoid contradictions and odd constructions.
- Be coherent. Have a meaningful flow of text.
- Be informative. Provide context with relevant information.
- Do not produce fake news.
- Have fewer than 280 characters (to meet Twitter's requirements).

In order to find the most appropriate approach, both extractive and abstractive summarisation algorithms were investigated. Four extractive summarisation algorithms were evaluated: (1) TextRank [45]; (2) Luhn's heuristic method [46]; (3) selecting the sentence with the highest emotional value for sadness and surprise; and (4) selecting the text in allegation (claim field) and evaluation (classification field) in the report. Five abstractive summarisation models were also tested: (1) GTP-2 (GPT-3 is the more robust version, as it uses a large amount of data in the pre-training phase; however, it was not used, as it is not available as open source) (gpt2-medium); (2) BART (facebook/bart-base); (3) BERT (bert-large-uncased); (4) XLNet (xlnet-base-cased); and (5) T5 (t5-base).

To analyse the results obtained by these models, one hundred news items were randomly selected from the dataset, and each of the previous approaches was applied. The results were assessed through automated and human evaluation. The former ensured that no fake news content was generated and that the maximum length of the post was respected. The human evaluation was carried out by means of a questionnaire given to a group of eight volunteers, who were asked to rate the coherence, consistency and informative quality of each post in binary terms in order to analyse the subjective side of the results. The verification of fake news was carried out using the tool Fake news classifier (https://fake-news-detection-nlp.herokuapp.com, accessed on 1 May 2022). This classifier uses the BERT model, which achieved 98% accuracy and 99% recall and precision during validation. See Table 2.

**Table 2.** Summary of the evaluation results. The em dash (—) means that the evaluation was not carried out because the approach had already been rejected.

| Approach | Size (N < 280) | True Context | Acceptance |
|---|---|---|---|
| T5 | 86% | 96% | 79% |
| BERT | 60% | — | — |
| BART | 100% | 94% | 25% |
| XLNet | 60% | — | — |
| GTP-2 | 60% | — | — |
| Luhn's heuristic method | 23% | — | — |
| Text-Rank-NLTK | 15% | — | — |
| Emotion-selection | 63% | 53% | — |
| Allegation sentence | 100% | 100% | 83% |

For extractive summarisation, methods based on Luhn's heuristic method and Text-Rank-NLTK were discarded because the maximum post size was not respected in most examples. The emotion selection approach was also abandoned because of the posts it generated, 37% were longer than 280 characters and almost half of them were classified as fake news (possibly because fake news is mostly associated with surprise and sadness). For abstractive summarisation, 95% of the summaries from BERT, GPT-2 and XLNet were the same. GTP-2 was chosen to represent the results of this group. However, the GTP-2 model was then discarded, as it led to meaningless sentences (96% of the generated tweets were irrelevant, without knowledge or information related to the fact-checking news).

The results show a greater acceptance of the T5 model and the Allegation Sentence Witch as approaches chosen to obtain the text-core element of the post. Although the T5 model had a 14% chance of generating a post with a size greater than 280 characters, the content produced had only a 4% chance of creating fake news. This model had a 79% acceptance rate among the study participants. The use of the allegation sentence, in addition to not producing fake content and the size always being within the expected range, has an acceptance rate of 83% in the human test carried out. On the other hand, the results of the BART model had a low acceptance rate among the volunteers in the study. The main reason is that in many cases, the generated sentence was not complete, and part of the message remained incomplete as shown in Figure 6a. This means that the size was respected, but the consistency and informative character were lost. Finally, Figure 6b presents the actual tweet on this subject posted by Snopes in Twitter for comparison.

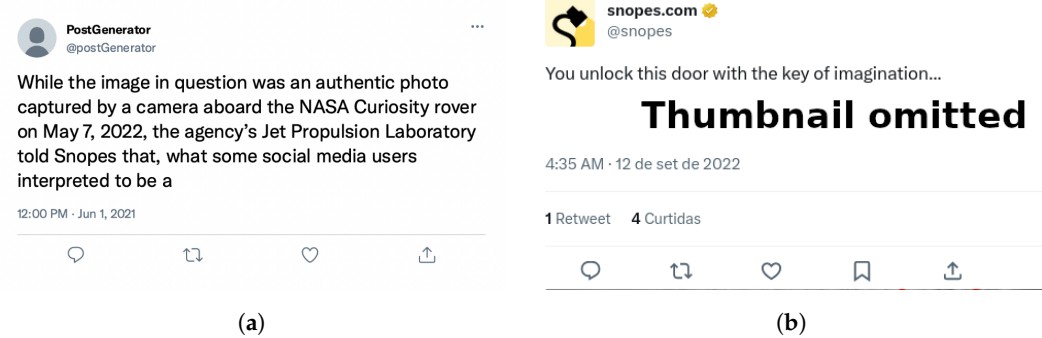

**(a)**          **(b)**

**Figure 6.** Tweet samples for reference. (**a**) BART model summarisation. (**b**) Actual Snopes tweet.

Note that the second and third level evaluations depicted in Table 2 are a bit more subtle than the first. In short, both T5 and BART are encoder–decoder models known for their state-of-the-art summarisation (with T5 outperforming BART in this task). The main architectural difference between T5 and BART lies in their masking technique. T5 uses

Masked Language Modelling (MLM), which involves masking words in the input text and training the model to predict the masked words. This forces the model to learn the relationships between words and phrases, even when some of the context is missing. This allows the model to adapt to different tasks without having to retrain the entire model. BART, on the other hand, uses a span-masking technique in which entire spans of text are masked, and the model is trained to predict the masked span. This technique is more effective for extractive summarisation because it forces the model to focus on the most important parts of the text.

Another important difference between T5 and BART is the training data. T5 is trained on a massive dataset called the Colossal Clean Crawled Corpus (C4), which contains over 1.5 trillion words of text and code. This large dataset provides the model with a broad understanding of language and helps it to generate more accurate and informative summaries. BART is trained on a smaller dataset that includes BookCorpus, OpenWebText and CNN/DailyMail. This dataset is less comprehensive than C4 but more focused on the types of text that BART is typically used on, such as news articles and blog posts. It is beyond the scope of this paper to delve into the explanatory power of these models, but considering only the training data, it is not unexpected that T5 outperformed BART in the online social media domain. In addition, the T5 model is fine-tuned to certain task-specific datasets, resulting in better nuance handling and improved performance.

The allegation sentence, in turn, is essentially the headline of the fact-check report. A news headline is a concise summary of a news article. Because its purpose is to grab the reader's attention and entice them to read further, it is a natural candidate for use in an online social media post. Headlines match the characteristics of engaging posts in the sense that effective headlines are clear, informative and engaging, using strong verbs, vivid language and relevant keywords. In other words, headlines are essential tools for communicating news effectively. They play a crucial role in attracting readers, summarising important information and making news articles discoverable online—all desirable characteristics of a good post. However, because appealing headlines are frowned upon, journalistic headlines tend to be as matter-of-fact as possible, which does not work in online social media.

In summary, the strength of T5 is that it is automatic, allowing the massive production of posts. Such a massive production would be suitable to feed a botnet and support repackaging to spread the same information several times. The weakness is that 4% of the tweets produced would be fake news instead of real content. The higher the scale, the higher the concrete number of tweets that spread false information as fact checking. This is a major drawback to using such technology to distribute fact checks. An alternative would be to use it in a cyborg context [47], where, for example, each production is filtered or adjusted by a human before being disseminated. The strengths and weaknesses of the accusatory sentence are reversed, losing in scale but gaining in quality.

### 3.2. Emotion Reinforcement

The strategy of emotion reinforcement is to add emojis and hashtags to the text-core as described in the last section, and to emphasise keywords.

Positive and negative emoticons, classified according to [48], are added to the tweet in different numbers, up to four, cf. [49], depending on the amount of space available (respecting the 280 character limit). The post is updated with emoticons before hashtags because they are more successful in increasing engagement rates [50]. Hashtags allow people to find tweets, especially if they are trending and relevant. By applying Latent Dirichlet Allocation (LDA) with Dirichlet-distributed topic word distributions to the text of the fact-check report, relevant topics can be extracted (skip the inter-document step) [51] and those with higher emotional levels are included in the post as a hashtag.

In addition, to increase the emotional appeal of the tweet, the most important aspects are highlighted with capital letters. This allows the user to quickly identify the content of the post, which can lead to greater engagement. To do this, the text-core of the post is

submitted to the KeyBERT [52] model to identify the keywords. This does not add any new information to the tweet.

For a reference, Figure 7 depicts an instance of applying the suggested reinforcements to a plain allegation extracted as described in the last section. Roughly, the emotion assessment is performed by a lexical approach cf. [53], based on the EMOLex dataset [54].

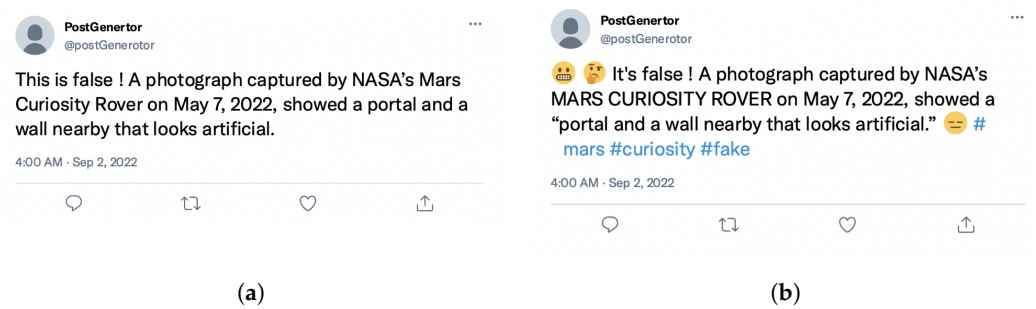

|         |         |
|:-------:|:-------:|
| (**a**) | (**b**) |

**Figure 7.** Emotion reinforcement instance. (**a**) Plain allegation sentence: `anger` 0.0; `anticipation` 0.0, `disgust` 0.0, `fear` 0.0, `joy` 0.0, `negative` 0.0, `positive` 0.0, `sadness` 0.0, `surprise` 0.0, `trust` 0.0. (**b**) Emotional reinforcement: `anger` 0.0; `anticipation` 0.0, `disgust` 0.0, `fear` 0.0, `joy` 0.0, `negative` 1.0, `positive` 0.0, `sadness` 0.0, `surprise` 0.0, `trust` 0.0.

### 3.3. Mental Process Reinforcement

The strategy for strengthening mental processes is based on replacing words in the text-core with synonyms associated with a particular mental process whenever possible. The approach is lexical, and the reference dataset is MENTALex [55]. See Figure 8 for an example.

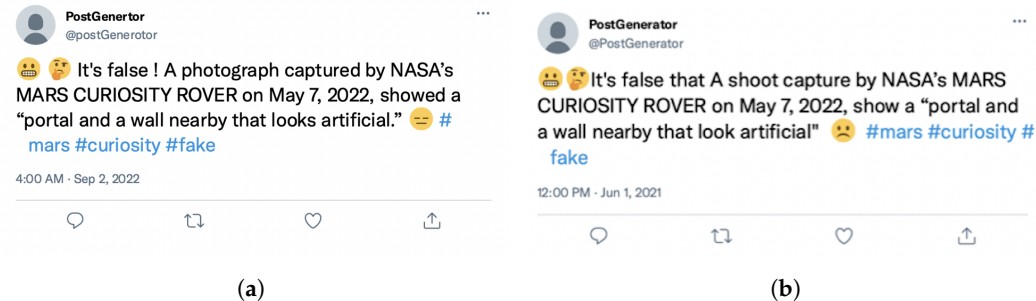

|         |         |
|:-------:|:-------:|
| (**a**) | (**b**) |

**Figure 8.** Mental reinforcement instance. (**a**) Emotion-reinforced sentence: `paranoid` 0.2, `neuroticism` 0.5, `schizoid` 0.4. (**b**) Neuroticism reinforcement: `paranoid` 0.0, `neuroticism` 0.9, `schizoid` 0.1.

### 3.4. Cognitive Reinforcement Assessment and Evaluation

In order to evaluate the results delivered by the prototype and, ultimately, to test the hypothesis of this paper, an experiment was conducted with 20 participants (the recommended number for a statically significant usability study [56]), between 12 and 18 September 2022. The anonymised volunteers were accustomed Twitter users recruited at the university, from whom the authors obtained informed consent. The experiment was conducted in a laboratory environment using the microblogging simulator [35]. The number of participants was chosen according to the guidelines suggested in [56].

The twenty participants are seven female and thirteen male, aged between 20 and 28. All the participants are Portuguese university students, studying computer science, public administration, bioengineering and psychology (unevenly distributed).

Participants were asked to interact with the platform in the same way they do when scrolling through their Twitter feed. The posts received voluntarily referred to ten news items. There was a total of 30 posts: 10 generated by the T5 model, 10 generated by the allegation sentence, and 10 posts extracted from the @snopes Twitter page. All posts were randomly presented to the user on the platform, without the user knowing which were generated by the prototype and which were from the @snopes Twitter page.

For reference, Figure 9 shows an evaluation instance. The same fact check produced by the three approaches under evaluation was presented to each user, who had the chance to engage with it or not. The results for these instances are also shown in the figure.

#prank eww It's false that all SIZES of Starbucks' PAPER CUPS hold the same amount of liquid. 🤢 💩

(**a**)

#fake #coffeeTime A fake video went viral on facebook that supposedly showed a person POURING COFFEE from one CUP into another that appeared larger. The video appeared to show that the same amount of COFFEE filled both CUPS to the brim 🤣 🤨

(**b**) T5

According to a popular internet rumor, Starbucks is scamming its customers with its trademark paper cup system, because, supposedly, all of its to-go cups hold the same amount of liquid.

Dan Evon reports. 👇

(**c**)

| Post | Likes | Replies | Retweets | Shares | Blocks | Follows |
|---|---|---|---|---|---|---|
| Figure 9a | 12 | 0 | 5 | 0 | 0 | 0 |
| Figure 9b | 11 | 1 | 4 | 0 | 0 | 2 |
| Figure 9c | 1 | 0 | 0 | 0 | 5 | 0 |

(**d**)

**Figure 9.** Example of a same post considering each strategy with results. (**a**) Allegation sentence. (**b**) T5. (**c**) Snopes tweet. (**d**) Engagement received for the presented posts.

Finally, the plot in Figure 10 shows the resulting engagement broken down by approach (T5 with cognitive reinforcement, allegation sentence with cognitive reinforcement, and actual Snopes tweets). In total, there were 434 interactions with the platform, an average of 21 interactions per participant and 14 interactions per post; in terms of interaction types, like was the most used by participants, followed by follow and retweet. This pattern is consistent with the one in [57].

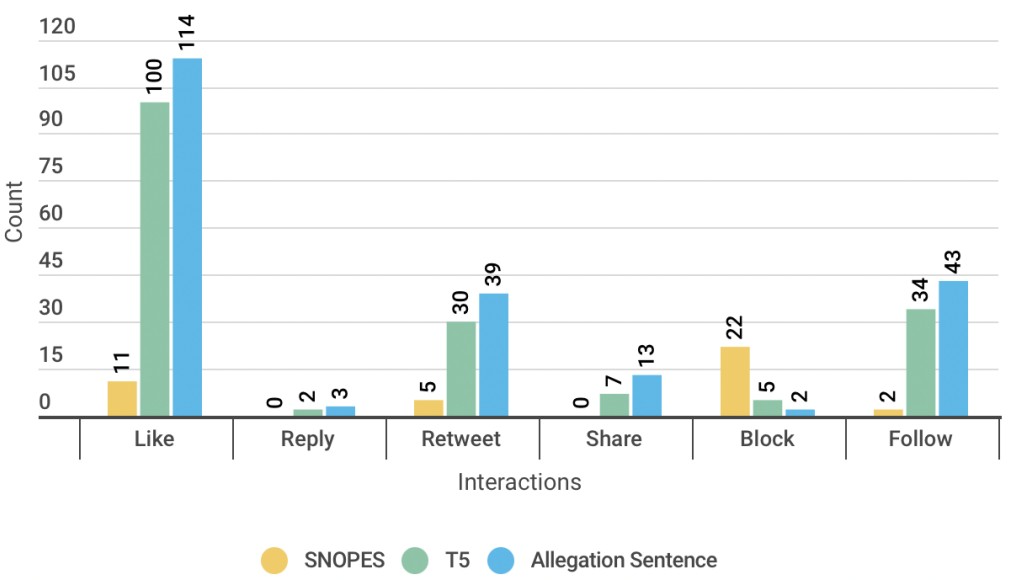

**Figure 10.** Interactions count per approach.

Note that in this study, unlike Twitter, the block interaction is associated with the post and not with the author's page. This means that when the participant blocks a post, it has a negative connotation towards the post. It takes an opposing viewpoint, and its use shows a lack of interest in the post.

Compared to the other approaches, engagement with actual Snopes tweets was very low and included most block interactions. See Figure 6b for a comparison with

Figures 7 and 8. Broadly speaking, Snopes tweets use more formal text and fewer emoticons and hashtags compared to the proposed heuristic.

As this study was conducted in a controlled environment, it is possible to use an appropriate measure of engagement through $\frac{interactions}{visualisations}$. The idea behind this metric is that the number of interactions with a post divided by the number of times that post is presented reveals how engaging it is. This is perhaps the simplest metric of engagement, but it is not often calculated, as most online social media do not provide the number of times a post has appeared. Assuming that each volunteer was exposed to the same number of tweets, the tweet produced by the prototype has an average engagement rate of 28% (for the T5 model) and 35% (for the allegation sentence). Therefore, an allegation sentence with cognitive reinforcement is the most appropriate heuristic, considering the issues addressed in this paper.

Despite the reduced sample size, it is nevertheless possible to make an informal assessment of these results using hypothesis testing and correlation analysis, at least for an indication. The values for the t-test comparing the Snopes results with both the allegation sentence and the T5 enhanced tweets, and the correlation between the enhanced (Snopes) and unenhanced (allegation sentence) samples are shown in Table 3. Although it does not provide compelling evidence for all the parameters measured, the results suggest that this is a path worth pursuing. The results for like may be explained by the fact that people tend to like posts more easily than the other actions, so a larger sample would also be needed, nonetheless, for a proper assessment.

**Table 3.** T-test and Pearson correlation values for the data presented in Figure 10.

| | **H: Snopes $\neq$ Allegation** | **H: Snopes $\neq$ T5** | **r: Snopes, Allegation** |
|---|---|---|---|
| Like | $-6.75, p = 2.497 \times 10^{-06}$ | $-5.44, p = 3.619 \times 10^{-05}$ | $0.85, p = 2.497 \times 10^{-06}$ |
| Reply | $-1.41, p = 0.176$ | $-1.50, p = 0.150$ | $0.31, p = 0.176$ |
| Retweet | $-3.13, p = 0.005$ | $2.82, p = 0.011$ | $0.59, p = 0.006$ |
| Share | $-1.89, p = 0.075$ | $-2.05, p = 0.055$ | $0.41, p = 0.075$ |
| Block | $2.20, p = 0.041$ | $1.63, p = 0.120$ | $-0.46, p = 0.041$ |
| Follow | $-2.44, p = 0.025$ | $-2.12, p = 0.048$ | $0.50, p = 0.025$ |

## 4. Fact-Check Profile

Before delving into the discussion on the presented results, it is necessary to understand how Snopes' fact checks spread on Twitter. In order to track the spread of fact-check reports, tweets containing a link to the Snopes website were searched for. The two main limitations of this search are that no network analysis was performed, and tweets that mention Snopes but do not provide a link to the website were not included in the sample.

For this analysis, two samples were collected, one "transversal" [2] and another "longitudinal" (the quoted terms are not used in the strict sense). The first, based on the Hot-50 page (the 50 most popular content on the Snopes website) on 20 July 2019, covers a period of 10 years (between 2008 and 2019) but with a smaller number of 50 fact-check reports. The second, comprising a larger sample but within a narrower time frame, includes 1512 fact-check reports published between 09/2019 and 09/2020. Note that this period is within the COVID-19 outbreak, which may introduce some bias. Then, for reference, Figure 11 shows the fact-check profile for ratings and Figure 12 for categories.

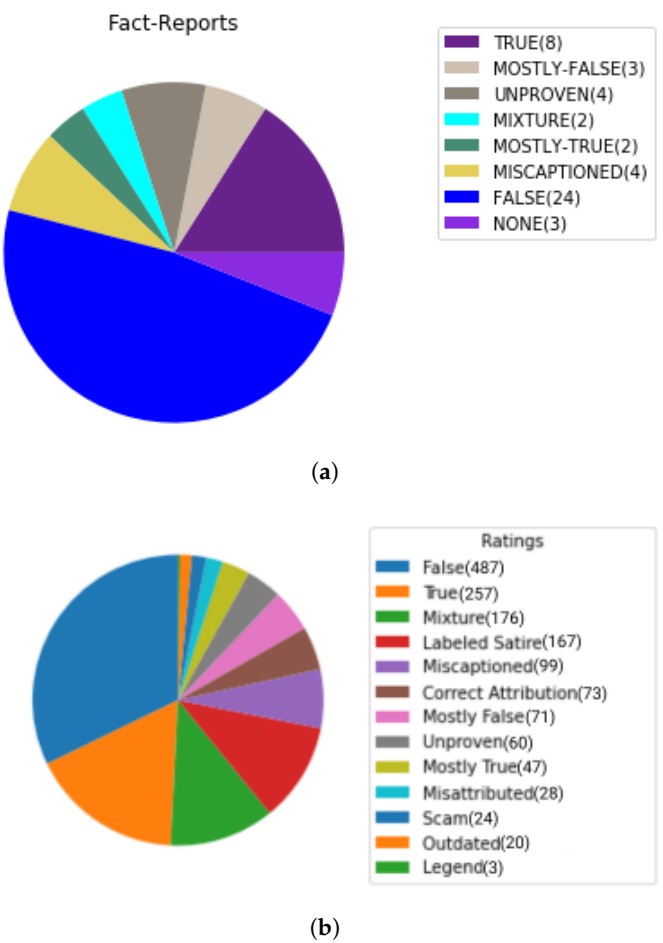

**Figure 11.** "Transversal" and "longitudinal" Snopes' ratings profile. (**a**) Transversal ratings. (**b**) Longitudinal ratings.

The analysis approach is based on both quantitative and qualitative analyses. The quantitative part is based on descriptive statistics, i.e., countable elements such as tweets, likes and shares. In addition to counting these elements and organising them in a convenient way, it also includes measures of central tendency and dispersion (mean, standard deviation, etc.) in order to present the behaviour that represents most of the tweets of this type. The qualitative part is mainly based on thematic analysis, which aims to identify patterns and themes in the data, but some narrative analysis is also carried out. In this sense, different groups of users are splatted (possible bots, Q1 spreaders, etc.), and their tweets contents are analysed and summarised in terms of the presented patterns. A combined analysis is also carried out in the search for textual patterns that are successful in online social media.

Snopes classifies its fact checks along two dimensions: ratings and categories. Ratings relate to the nature of the claim, i.e., true, false, and mix, and categories relate to the subject of the claim, i.e., politics, medicine, and junk news. Note that some events are the same in both datasets, for example, the higher number of fact checks in both cases is related to false politics news. Given that some of the proportions are maintained in both surveys, it is possible to suggest the existence of a pattern that needs to be explored in future work. Nevertheless, it is possible to suggest that the data are coherent.

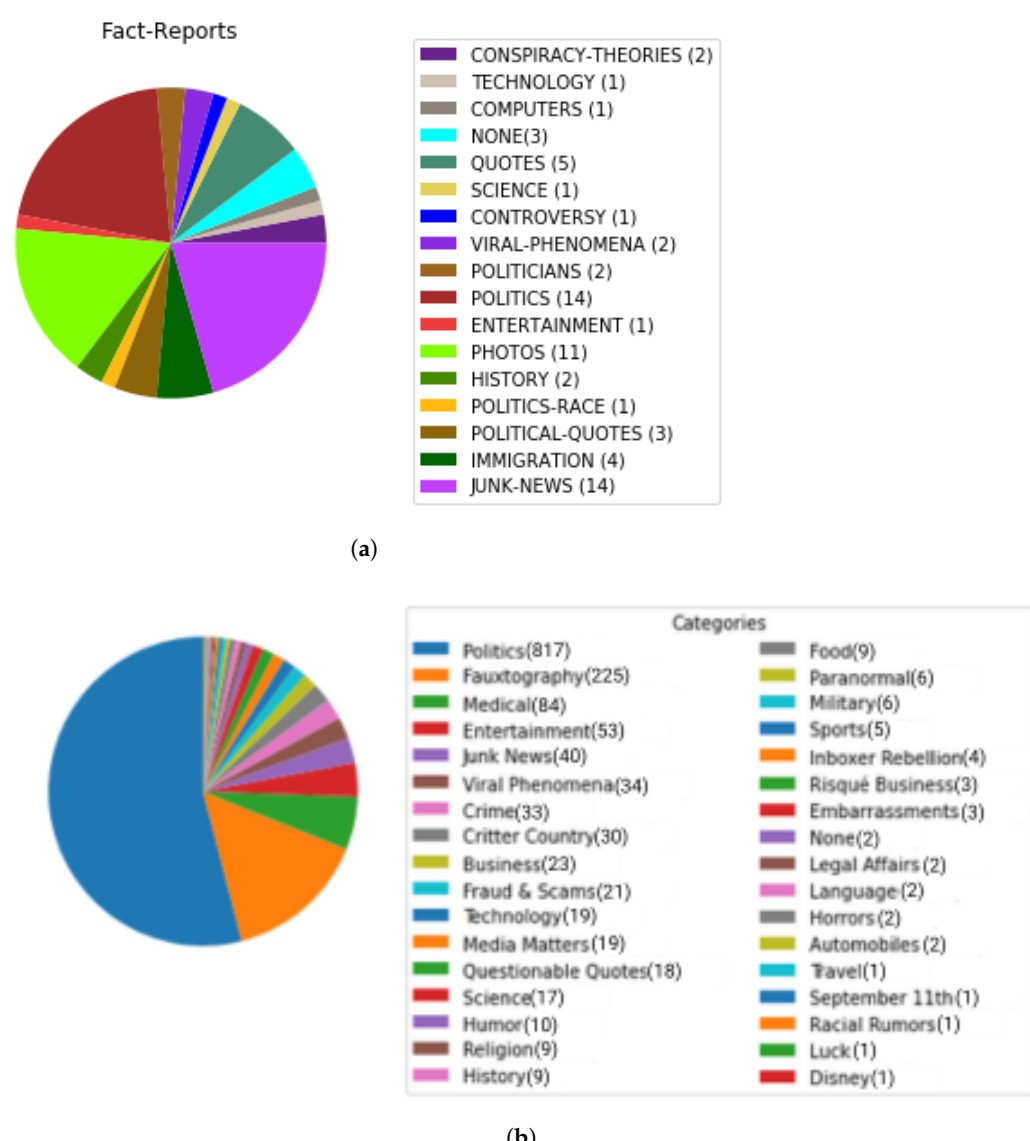

**Figure 12.** "Transversal" and "longitudinal" Snopes' categories profile. (**a**) Transversal categories. (**b**) Longitudinal categories.

For each fact check collected in the longitudinal dataset, all tweets containing a link to the fact-check report were collected using the Twint scrapper (https://github.com/twintproject/twint accessed on 1 July 2019). The tweet base is shown in Table 4. Assuming that Snopes is sufficiently representative, we can say that the spread of fact checking is not doing well (corroborated by [7], who claim that well-executed fact checking is an exception). The number of 120,026 tweets results in an average of 79 tweets per fact-check report, a negligible figure. It is even worse when you consider that this is a sample of one year's worth of fact checks, so on average there are about 10,002 tweets spreading Snopes' fact checks per month; for reference, 500 million tweets are sent daily [58]. Note that deleting tweets is part of Twitter's dynamic [59]. Therefore, some tweets may be posted shortly after the fact check is published and deleted before this collection, especially for fact checks published several months before the data collection. Similarly, some of the more recently published fact-check tweets may have been deleted after this survey. The impact of deleting tweets is not assessed in this paper, but it is not expected to change this ratio significantly.

**Table 4.** Snopes fact checking spreading on Twitter (global level).

| Total Tweets | Unique Author | Replying Tweet | Unique Mentions | Tweets w/ Mention | Unique Hashtags | Tweets w/ Hashtag |
|---|---|---|---|---|---|---|
| 120,026 | 70,173 | 93,438 (78%) | 224,461 | 93,461 (78%) | 18,340 | 7799 (6%) |
| **No. of Replies** | **Replied Tweets** | **No. of Retweets** | **Retweeted Tweets** | **Number of Likes** | **Tweets w/ Likes** | **Hour (Top3)** |
| 76,630 | 36,158 (30%) | 194,073 | 16,414 (14%) | 522,228 | 39,457 (33%) | 16 h, 21 h, 22 h (6% each) |

However, it is worth expanding on Figure 4 and then referring to Table 5 for showing data at the fact-check level, e.g., how many tweets mentioned a given fact check, and at the tweet level, e.g., how many likes a fact-checking tweet received.

The distribution of fact checks is shown in Table 5 (and in Figure 13). As mentioned, on average, each fact check is spread by $79 \pm 388$ tweets, the maximum number of tweets for a fact check is $\approx$13 k, and the maximum number of retweets is $\approx$30 k, so no fact check in the sample "went viral". However, the data show that some fact checks attract more engagement than others, but in addition to being a rare event, as it was not possible to find a reasonable explanation, the cause is considered an ad hoc event. Also, on average, a fact check is shared by $11 \pm 3$ people, a rather low number, suggesting that fact checks do not have a broad audience impact, which may be related to individual interests. On average, a fact check is shared for 37 days after its publication, but in some cases it is shared for more than a year. The first tweet about a fact check is also sent, on average, one day after it is published, but it can remain dormant for several months, even taking into account Snopes' Twitter profile. Note that none of these tweets went viral; the number of likes received by the tweet with the most likes is insignificant compared to the engagement received by viral fake news.

**Table 5.** Descriptive statistics for fact-check spread profile on Twitter.

| **a. Fact-Check Level** | | | |
|---|---|---|---|
| Fact Check | AVG | Std. D. | Max | Min |
|---|---|---|---|---|
| TWEETS | 79 | 388 | 12,787 | 0 |
| AUTHORS | 11 | 3 | 15 | 3 |
| MENTIONS | 92 | 399 | 9931 | 0 |
| HASHTAGS | 7 | 27 | 912 | 0 |
| REPLIES | 51 | 244 | 6485 | 0 |
| RETWEETS | 128 | 1019 | 30,489 | 0 |
| LIKES | 345 | 2274 | 66,002 | 0 |
| DAYS ($\Delta$) | 37 | 65 | 496 | 0 |
| DAYS ($t_0$) | 1 | 8 | 153 | 0 |
| **b. Tweet Level** | | | |
| per Tweet | AVG | Std. D. | Max | Min |
| MENTIONS | 2 | 3 | 50 | 0 |
| HASHTAGS | 0 | 1 | 36 | 0 |
| REPLIES | 1 | 10 | 2374 | 0 |
| RETWEETS | 2 | 70 | 16007 | 0 |
| LIKES | 4 | 150 | 34464 | 0 |

**Table 5.** *Cont.*

| c. Author Level (Q1–Q3) | | | | |
|---|---|---|---|---|
| Author Q1-3 | AVG | Std. D. | Max | Min |
| Fact Checks | 27 | 108 | 1339 | 1 |
| Tweets | 70 | 261 | 3521 | 22 |
| Mentions | 1 | 3 | 49 | 0 |
| Replies | 1 | 22 | 2374 | 0 |
| Retweets | 6 | 52 | 3308 | 0 |
| Likes | 13 | 98 | 6561 | 0 |
| Hashtags | 0 | 1 | 11 | 0 |
| Followers | 27 k | 300 k | 4.1 M | 1 |
| Following | 2704 | 9129 | 117 k | 0 |

The data presented show that fact-checking reports, at least from Snopes, have a negligible presence in online social media, at least on Twitter. Note that finding aligns with the result obtained in Figure 10.

For a qualitative profile, it is worth targeting groups of profiles of interest (the handles are anonymised for privacy reasons).

The first group of interest is the probably bots group. They are called probably bots because of the limitations of the Botometer [60]. However, it is assumed that these are automated accounts. In this sense, on the one hand, it is a pleasant surprise to find automated profiles spreading fact checks, but on the other hand, most of these profiles, as far as can be assessed, are not sophisticated and, in addition, some are probably cyborgs (partially automated accounts). It should be noted that most fake-news spreaders are also cyborgs [61], whose automation is focused on inflating the spread [62] and faking widespread consensus [63].

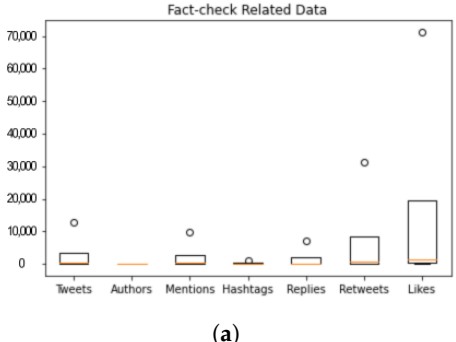

(a)

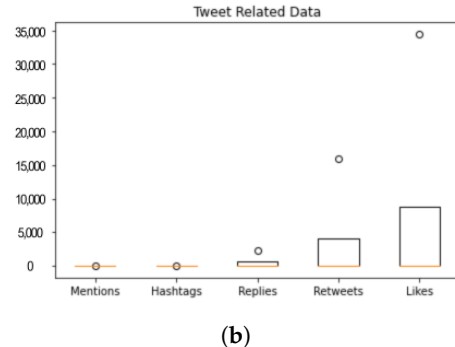

(b)

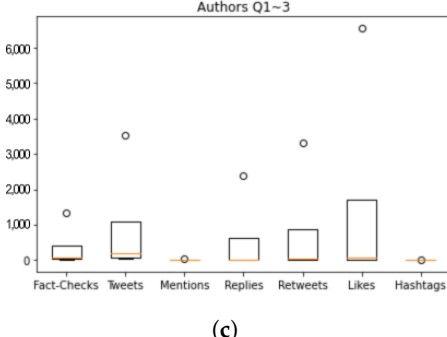

(c)

**Figure 13.** Visual presentation of descriptive statistics depicted in Figure 5. (**a**) Fact-check level. (**b**) Tweet level. (**c**) Author level (Q1–Q3).

Perhaps the most successful account in this group is @0002 (16.4 k followers), which uses two strategies, one of which is to tweet newspaper titles and headlines with the corresponding link as the text of a tweet (see Figure 14). It performs sentence tokenisation

on the headlines and selects as many complete sentences as possible, constrained by the 280 character limit (excluding title and link). As it removes the broken lines, it appears to be an opinion taken from the news. Most tweets are first (non-replying) tweets from the Washington Post. Other strategies include tweeting raw headlines, using the share button on agency websites and curation services such as Scoop.it (https://www.scoop.it/, accessed on 14 November 2023), retweeting, etc.

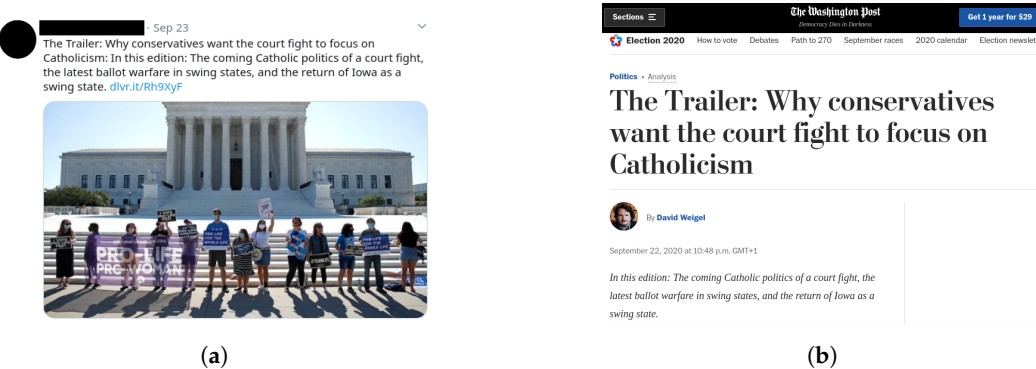

(**a**)                       (**b**)

**Figure 14.** Example of an automated account. (**a**) The @0002 tweet. (**b**) The respective Washington Post news.

Another group of interest is fact-checking agencies. Thus, to compare with @snopes, some other fact-checking agencies were searched on Twitter (refer also to Table 1 and Figure 1). Roughly, the @snopes behaviour is based on sending a fact-check link but with a new and possibly funny headline for the tweet text. It will also send the same fact check more than once, with a different headline for the text, which may just be the news rating, such as "× False". For retweeting, it uses the strategy of sharing tweets with links to its portal (it was not possible to establish whether there is any curation). It is not common for @snopes to reply to other people's tweets.

@PolitiFact (joined in 2007, 673.6 K followers), @Poynter (2007, 214.8 K) and @factcheckdotorg (2009, 190.3 K) all show similar behaviour, the social articulation of which varies according to the style of the person writing the tweets. @Channel4News (2008, 99.7 K) also shows the same behaviour but participates more in replies and some other Twitter interactions. Despite this, this profile has fewer followers compared to the others. Note, however, that @Channel4News followers were retrieved after almost a year (in May 2021) and it grew to 2.4 million followers, while the others followed the expected growth rate.

@APFactCheck (2011, 31.5 K) is a bit more successful in spreading than the other agencies; its tweet texts are more incendiary than journalistic, perhaps better suited to Twitter. For a reference, see Figure 1. Although @APFactCheck seems to be more successful in spreading than the others, it is less popular compared to the number of followers. This leads the discussion to the popularity and engagement of a profile. A more balanced journalistic tone tends to be more popular, while a more "tabloid" tone tends to be more engaging.

Nevertheless, proper social behaviour is essential as shown by the @Channel4News. This finding is consistent with the journalistic view of the role of emotion in news [64]. Finally, @MBFC_News (joined in 2015, 4411 followers) and @TheDispatchFC (2019, 1277) only tweet the title of the fact check and its link. This suggests that sharing a title and a link is not a good strategy for disseminating fact checks on Twitter.

The third group of interest comprises the top spreaders (considered only the Q1 group). Leaving aside the bots and agencies already mentioned, the behaviours worth highlighting for this section are those of @0016 (1029 followers, joined 2008, 2040 tweets), @0005 (14.3 K, 2012, 193.6 K), @0014 (56, 2020, 1292), @0008 (34, 2014, 3251), @0017 (58, 2020, 5068), @0015 (2321, 2016, 37 K) and @0009 (149, 2013, 13.2 K) sending the same fact-check link a hundred times, mostly in response to a particular sensitive topic. This suggests an emotional response from the user and reinforces the importance of emotion in spreading as already known from the fake news phenomenon and journalism [64]. Despite this peculiarity, they usually

behave like the previous profiles. Although @0009 fits with these profiles, they act as activists by mentioning people and including calls to action in their tweets. Such a strategy requires the profile to be able to respond to such engagement by showing appreciation [65]. Figure 15 presents the relation between spreading and fact checks.

Note that Twitter is used during people's leisure time, mostly for fun. It is therefore natural that standard text does not attract attention. Also, in almost half of the tweets evaluated in [2], the tweet consists only of the link or similar plus the fact-check title. Finally, the profiles @0006 (3380 followers, joined 2009, 15.9 K tweets), @0007 (135, 2019, 11.9 K), @0018 (74, 2013, 11.2 K), and @0019 (34, 2018, 5364) are mostly focused on replying. Some are more "polite" or technical, matching the journalistic form, while others are more "passionate", matching the tabloid form. This also supports the idea of differences and uses for restrained and enthusiastic tweet texts.

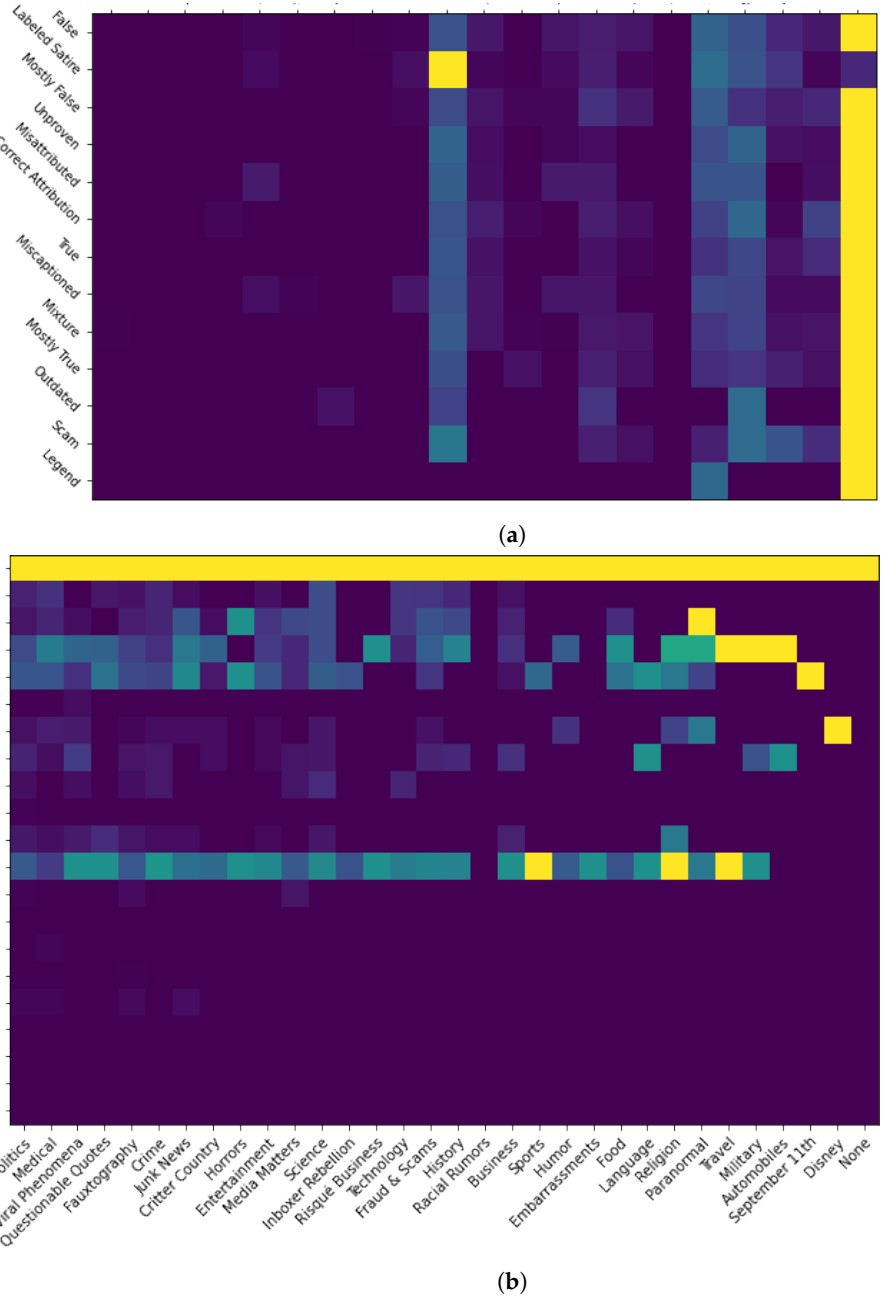

(**a**)

(**b**)

**Figure 15.** Heatmaps relating Q1 authors and fact checks; refer to Figures 11 and 12. The dark blue area is due to the volume difference between the spreaders that ranges from 83 to 3521. (**a**) Ratings. (**b**) Categories.

It is also worth noting that the dispersal dynamics of fact checks do not seem to follow the dispersal behaviour of fake news [62]. Figure 16 shows that despite the initial concentration of dispersal just after publication by the fact-checking agency, fact-check reports, on average, retain their relevance within the year with "late" first dispersals. Based on the graph, one could argue that Q1 authors perform the initial spreading, and Q2 and Q3 authors amplify the spreading. However, the number of retweets for Q2 ($1 \pm 7$ [0–112]) and Q3 ($1 \pm 7$ [0–340]) is too low to support this idea. Presumably, people in Q2 and Q3 are looking for the fact check to reply to, rather than responding to actual sharing behaviour.

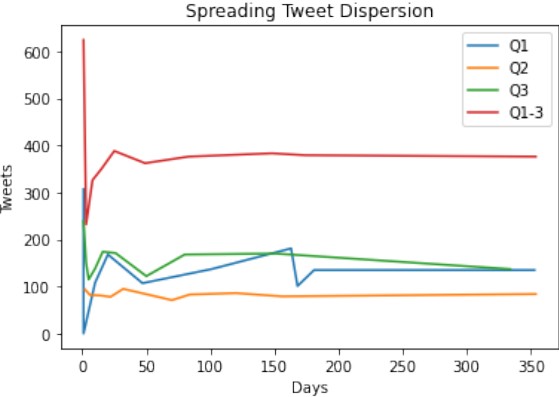

**Figure 16.** Spreading tweet dispersion.

Overall, the findings suggest that a fact-checking profile should behave like a normal tweet user in order to increase distribution.

Finally, the common silence of official sources towards fake news is a bad practice due to the agenda-setting phenomenon [66]. Agenda-setting, in a nutshell, is the ability of the mass media to determine the broad issues discussed by the general public. Most of the official bodies attacked by fake news remain silent, believing that it is not worth responding to absurdity. However, this is a bad practice because fake news is also able to set themes, and without a response, it ends up being the only narrative. If a lie is sown properly and for a long enough time, it can be taken as truth by the general public. Therefore, a response from credible and official sources is necessary. From this perspective, a boosting strategy would be to use the mediation algorithm to weave tweets with fact-checking content into the timeline of people who consume fake news. Such a strategy would be more beneficial than shadow banning content.

## 5. Discussion

Millions of users use posts to share their thoughts, opinions, news and personal information. In social media studies, the content of posts is often used as a basis for research, as it provides insight into public opinion and what people are talking about [57]. Natural language processing (NLP) techniques transform the content of posts into data that can be interpreted by a computer. An example is the use of a GPT-2 and the plug-and-play language model to incorporate an emotion into the output of a generated text, ensuring grammatical correctness [58]. It is also possible to use a multiagent approach to engaging Twitter users with fact checkers [59]; the idea is to encourage users to share verified content by educating Twitter users sharing URLs already flagged as fake [60].

The surge of fake news and misinformation, especially on platforms like Twitter, necessitates rigorous fact checking. However, even within this realm, there is a prevalent bias with a predominant focus on Western platforms, side lining major Eastern social networks. While several strategies have been proposed to counteract the spread of misinformation, a glaring gap persists: the absence of emotional and personality considerations in fact-checking posts. These elements are instrumental in enhancing reader engagement, resonating with the current trends of digital marketing. This document looked for bridging

this gap by emphasising the role of emotions and personality in the presentation of fact-checking data, aiming to ensure that truth not only reaches the masses but also resonates with them.

The exploration of the dissemination landscape of Snopes' fact checks on Twitter provides an insightful reflection on the multifaceted nature of information spread in the digital era. The research delineated two disparate analysis cohorts: a "transversal" ensemble spanning a decade with a modest compilation of 50 fact-check reports, and a "longitudinal" assemblage encompassing a year with a substantial tally of 1512 reports. It is through this nuanced examination that the study unveiled the categorical dichotomy of Snopes' fact checks: an epistemic bifurcation into ratings encapsulating the veracity spectrum (true, false, and mix), and thematic compartmentalisation into subjects like politics and medicine. The research unveiled the stark reality of digital dissemination; the meagre average of 79 tweets per fact-check report paints a sombre picture of the virality abyss that fact-checking endeavours seemingly plummet into.

For an extrapolation, it is possible to consider the ratio between the result shown in Figure 10 and Table 5, shown in Table 6. In short, it first calculates the ratio of interactions received on Snopes' posts to the other approaches, e.g., snopes:T5. From Figure 10, the ratio of like is 11:100 ⇔ 1.1:10. This means that for every like received by a Snopes tweet, it can be expected to receive approximately 10 like using the T5 reinforcement. Multiplying this number by the average number of likes received by a Snopes tweet, as shown in Table 5b, which is 4, $4 \times 10$ can be used to suggest that T5-reinforced tweets can be expected to receive an average of about 40 likes.

**Table 6.** Engagement extrapolation from Figure 10 and Table 5b.

| Action | T5 | Allegation Sentence | Extrapolation (T5) | Extrapolation (A.S.) |
|---|---|---|---|---|
| like | 1.1:10 | 1.1:11.4 | $4 \times 10 = 40$ | $4 \times 11.4 = 45.6$ |
| reply | 0:2 | 0:3 | $1 \times 2 = 2$ | $1 \times 3 = 3$ |
| retweet | 1:6 | 1:7.8 | $2 \times 6 = 12$ | $2 \times 7.8 = 15.6$ |

Note that the presented extrapolation is linear; if an exponential extrapolation is used (more engagement means the post is presented to more people, resulting in exponential engagement), it is possible to assume that these figures would be slightly better. For example, if the average number of likes on a tweet is 80 [58], then the linear extrapolation is still below the Twitter average; considering an exponential extrapolation, it is possible to suggest that it could reach this threshold. Considering the result in Table 5a, this would increase the average number of likes from 345 to 3160 (considering the linear extrapolation).

As a result, it is not possible to argue that this strategy alone would solve the problem of fact-check propagation, as the numbers are still small given the Twitter ecosystem, but it is possible to argue that an artefact as such would be a component of such a solution.

Further on, the discussion navigated the digital milieu of Twitter, unfolding the trinity of principal actors steering the dissemination narrative: automated personas, fact-checking agencies, and the fervent top spreaders. It also highlights the digital comportment of renowned fact-checking stalwarts like @PolitiFact, @Poynter, @factcheckdotorg and the emblematic @snopes, albeit with variegated levels of engagement. An important revelation emerges from the section, sketching the emotive response of individual accounts, whose repetitive sharing of the same fact-check link, predominantly as a retort to sensitive topics, unveils a tapestry of emotional user engagement. For a reference, Figure 17 shows the valence of the tweets sampled (only valence is shown, as it was the reinforcement target; refer to Figure 7). Note that most of the tweets are neutral, suggesting a "journalistic style".

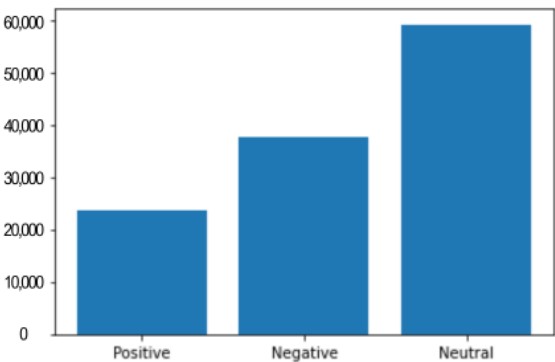

**Figure 17.** Sum of valences calculated using VADER; it was excluded from the sample tweets composed only by links, title, hashtags, and mentions via the @snopes tag.

The text ventures further, positing a compelling supposition that perhaps the digital demeanour of fact-checking profiles should echo the habitual tendencies of a quintessential Twitter user to amplify dissemination. This notion resonates with the broader approach that underscores the indispensable role of emotional and personality nuances in crafting a more engaging fact-checking discourse. Yet, amidst this discussion, the text addressed the onus that rests upon the shoulders of official sources. The suggestion of a mediation algorithm, envisaged to seamlessly interlace fact-checking content within the digital timelines of fake news consumers, emerged as a promising antidote, juxtaposed against the less efficacious shadow banning tactic.

The article culminated in a holistic reflection, subtly nudging towards a paradigm shift in combating misinformation. It beckons a more profound understanding and integration of emotional and personality dimensions in crafting and disseminating fact-checking content. This meticulous exploration, entwined with pragmatic suggestions, heralds a hopeful stride towards not merely reaching the digital masses but resonating with the cognitive and emotional fabrics of the global digital populace. This discussion embodies a synthesis of technological finesse and a profound understanding of human interaction, endeavouring to foster a more informed and engaging digital discourse amidst the turbulent seas of misinformation.

Building upon the discussed framework, the integration of newly born technologies like Bard, LLAMA and GPT-4 can significantly bolster the effectiveness and reach of fact-checking challenges. For instance, the refined natural language processing capabilities of GPT-4 can be harnessed to develop a more nuanced mediation algorithm that not only interlaces fact-checking content within digital timelines but does so in a manner that is contextually relevant and emotionally resonant. By analysing the behavioural patterns and engagement metrics of users, alongside the prevailing discourse, GPT-4 can tailor the presentation of fact-checked information to align with the cognitive and emotional inclinations of the digital populace. This technological intervention, rooted in large language models and NLP, aims to craft a more informed and engaging digital discourse, thus navigating the turbulent seas of misinformation with a compass of attention and intervention, resonating with digital masses and helping on a level that fosters a culture of dialogue. These technologies are the main objects for future research.

*Broader Implications*

Citizens and civil society use both conventional and digital media to find out about reality. In the analogue context, such as the spread of disinformation in other periods of peak information intensity such as during the COVID-19 pandemic, a study by the Reuters Institute at the University of Oxford warns that between March and April 2020, up to 85% of false news in Spain came through online media, including social media [67].

There is also a notable increase in the use of news platforms and social networks by certain political actors and civil society, in some cases in response to the spread of content

that is closer to propaganda techniques than to real events. Benkler in [68] presents the political media landscape of the United States. The book focuses on the current epistemic crisis in political communication, highlighting the 2016 US presidential election. The authors present a map of the American media landscape. It is the result of an analysis of millions of social media posts and reveals a highly polarised and asymmetric ecosystem. The detailed case studies also trace the emergence and spread of disinformation in this context. As a result, the authors argue that the current problems of media and democracy are a consequence of an asymmetric media structure driven by political powers and their political agendas. Although technology is not responsible for the political crisis, it is playing an increasing role in the political information setting through algorithmic mediation.

For example, the social media that have a mediation algorithm, such as Twitter, have made efforts to curb disinformation campaigns like those promoted by Donald Trump. But there are also social media without mediation algorithms. A key issue with Telegram (and similar apps) is that, unlike other social media, Telegram does not have a mediation algorithm and, unlike WhatsApp, does not limit the number of users within a group. Therefore, all messages reach all subscribers as a broadcasting station. In other words, if Trump had used such a platform, his tweets would have reached even more people, and the Capitol incident would have potentially been even worse. This is why Telegram is becoming a trend [69]. As another example, to avoid YouTube strikes or censorship in their videos, the author can publish as private and then spread the links through Telegram [69], potentially reaching even more people. This mechanism is already being used, at least in Germany [70] and Brazil [69]. The disadvantage of these platforms is the huge number of messages, which is what motivated the mediation algorithm in the first place.

The emergence of new platforms such as Gettr raises concerns about the creative ways in which promoters of false content operate, creating new and serious problems with regard to the verification, contrast and veracity of content and information. Another example is what is happening with Telegram, which "is becoming the latest source of viral disinformation. It is harder to track how information travels within messaging apps and private conversations" [67].

The MediaLab of ISCTE, Instituto Universitário de Lisboa, focuses on monitoring advertising and disinformation on social networks. In a report on the 2019 Portuguese legislative elections, in partnership with Democracy Reporting International, it identified more than six thousand five hundred publications on Facebook with disinformation content or fake news. More than one million Portuguese had contact with "disinformation" and fake news content in the month before the elections [71]. In fact, the phenomenon is expected to increase in the next Portuguese elections in January 2022.

An example of a political disinformation agenda on digital social networks can be seen in the 2018 Brazilian elections [72]. Based on information from the Facebook Papers, the local media found a disproportionate influence of a group of accounts and pages responsible for most of the production of political content on Facebook. The actions of the so-called "super producers" of content and disinformation can be seen in the numbers. On the day of the first round of the presidential election, 18.4 million political publications were created by 6.7 million profiles or Pages on the platform; 35% of this material was published by only 3% of the accounts. This equates to 6.4 million political posts generated by just 201,000 accounts. Seventy-four million different people were exposed to the material, generating 2.74 billion views. Facebook researchers estimated that these accounts received 28% of these views, or 767.2 million.

A glimpse of what happened in Germany is presented by [70]. In short, it followed the structure of half-truths and aggressive language. But there was no evidence of a coordinated attack, and they did not gain traction. It followed the trend of using YouTube to gather supporters from different social networks and direct them to Telegram. Nevertheless, all parties contested the social media space (with different emphases and strategies) but with negative campaigns (aimed at attacking other parties rather than discussing their own proposals). The report highlighted that Alice Weidel received little attention despite being

a controversial candidate. This shows that it is not how controversial a candidate is—there are plenty of them in any election—but how well he or she is able to manipulate public opinion and spread their message through the population with the support of technology.

The media space for public discussion, political debate and access to informational reality is increasingly being built in the digital sphere, i.e., mediated by algorithms. It is an "uncanny valley" to establish a common normative framework between different cultures and countries that determines what is true or false and what is real or imagined. Contemporary initiatives seem to be built on the basis of some interests, and the result is the configuration of an agenda of issues that have more to do with the algorithms of social networks and their biases than with social priorities of public and general interest, respecting different societies and their cultures and values. To add to the complexity, "under human rights law, even the expression of false content is protected, albeit with some exceptions" [73].

Therefore, the digital social media landscape is undergoing relevant changes because "the digitisation of information is leading to the segmentation of audiences according to their tastes, preferences and interests" [74]. At the same time, the reinforcement of the situation, the use of native digital media and the greater accessibility of news through online channels (the Geiger study, 2019, revealed that in 2018, about four out of ten Americans received news through social media) are transforming and establishing users (audiences) into another constructor of the mediatic agenda [66].

## 6. Conclusions

The digital landscape, dominated by social media platforms such as Twitter, has witnessed the exponential rise of misinformation and the subsequent challenges associated with fact checking. This research ventured into a novel area by highlighting that by intertwining emotional and personality nuances within tweets, engagement with fact-checking tweets can be significantly increased.

To test this hypothesis, a set of heuristics was proposed, resulting in a prototype. The results of this prototype were then submitted for human evaluation in a laboratory environment.

Among the models studied, the T5 model and the allegation sentence produced the best results (also avoiding clickbaits), which were then used to build the prototype. According to the literature, the use of emoticons and hashtags helped to strengthen the emotional dimension. Furthermore, based on the Adaptive Personality Theory, the reinforcement of the neuroticism process helped to improve the overall result. Therefore, according to the data presented, the hypothesis was confirmed with an increase of 35% in the engagement score.

The prototyping phase, which focuses on human evaluation in a controlled environment, reaffirmed the importance of social media content presentation. While factual accuracy remains paramount, the manner in which content is presented, full of emotional undertones and psychologically relevant cues, is equally important. Tools such as KeyBERT, the application of LDA and the strategic use of emoticons and hashtags were instrumental in achieving the initial goals.

The study of disinformation across different digital social media platforms and their different algorithmic mechanisms highlights the complex and multifaceted nature of this issue. It points to the wider implications of the findings, which may extend beyond the specific context of Twitter. As shown in the analysis, each social media platform, whether algorithmically driven like Twitter and Facebook or less mediated like Telegram, plays a unique role in the spread and amplification of disinformation. This diversity in platform mechanisms suggests that strategies to combat disinformation must be tailored to the specific characteristics and user dynamics of each platform. Moreover, the different cultural contexts and types of disinformation found on different platforms and in different regions further underscore the need for a nuanced approach. By acknowledging these different contexts, this study aims to provide a basic understanding of how misinformation spreads in digital spaces that can be adapted and applied to different social media environments and

cultural settings. This adaptability is essential for developing effective, context-sensitive strategies to mitigate the spread of disinformation globally, in line with the growing need for a more comprehensive and culturally aware approach to tackling this pervasive problem.

In terms of limitations, it should be noted that the results presented in Section 3 do not fully represent all users of online social media but only the subset limited by demographic data (young adults studying at the same Portuguese university). Furthermore, although the data presented in Section 4 are sufficiently comprehensible, they are limited to one fact-checking agency (Snopes) and one online social media (Twitter), issues to be addressed in future work.

Future endeavours should dissect the heuristic presented, examining each feature as a distinct variable and optimising them individually for maximum impact. This will ensure a nuanced understanding and refined application of each element. In addition, a deeper psychological analysis of the generated content is crucial. This will not only validate the emotional depth of the tweets but also highlight areas that need fine tuning. Furthermore, comparative analysis with other social media platforms is needed to understand whether the findings are unique to Twitter or more broadly applicable. To be also included is how the presented proposal might adapt to evolving social media landscapes and misinformation tactics. As we move forward in the age of digital misinformation, it is vital to use every tool at our disposal to ensure that fact checking not only informs but resonates deeply with audiences.

**Author Contributions:** Conceptualization, F.S.M., A.d.C.O.S.G., J.J.A. and P.N.; Software, M.A.B.; Validation, J.J.A. and P.N.; Investigation, F.S.M.; Data curation, M.A.B.; Writing—original draft, F.S.M. and A.d.C.O.S.G.; Writing— review & editing, F.S.M., A.d.C.O.S.G., J.J.A. and P.N.; Supervision, J.J.A. and P.N. All authors have read and agreed to the published version of the manuscript.

**Funding:** This work is financed by National Funds through the Portuguese funding agency, FCT—Fundação para a Ciência e a Tecnologia within project 2022.06822.PTDC.

**Informed Consent Statement:** Informed consent was obtained from all volunteers involved in the study. In addition, all collected data are de-identified for privacy purposes.

**Data Availability Statement:** Data are contained within the article.

**Conflicts of Interest:** The authors declare no conflicts of interest.

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
