# Peer review of "Emotional and Mental Nuances and Technological Approaches: Optimising Fact-Check Dissemination through Cognitive Reinforcement Technique†"

_electronics, doi:10.3390/electronics13010240_

Round 1

Reviewer 1 Report

Comments and Suggestions for Authors

The present study is well etablished and written well. Here are some issues that require some attention. 

1. In section 3.4, there were 20 participants, but what was their mean age, gender distribution, and other demographic information? 

2. Please enlarge the figure 11. I can not read the letters. 

3. The authors should discuss why T5 and allegation sentence were powerful in more detail. 

Comments on the Quality of English Language

The English is acceptable. 

Author Response

Thank you for your insightful feedback on our manuscript. We have sought to follow your considerations carefully. The responses are below, and the effects in the paper are highlighted by the orange notes.

1. In section 3.4, there were 20 participants, but what was their mean age, gender distribution, and other demographic information?

Additional information has been included.

2. Please enlarge the figure 11. I can not read the letters.

For proper rendering, figure 11 is split into two, one for the rating, another for the categories.

3. The authors should discuss why T5 and allegation sentence were powerful in more detail.

Additional discussion has been included.

Reviewer 2 Report

Comments and Suggestions for Authors

Electronics-2748687-

Emotional and Mental Nuances and Technological Approaches:Optimizing Fact-Check Dissemination through Cognitive Reinforcement Technique

 This study highlights the multifaceted nature of fact-checking, considering technological, emotional, and public perception aspects. While content analysis through Natural Language Processing (NLP) has been instrumental in deriving insights from posts, various fact-checking entities on Twitter, such as @snopes, @PolitiFact, and @APFactCheck, adopt diverse strategies to disseminate verified content. Nonetheless, there remains a gap in incorporating emotional and personality nuances in fact-checking posts, which could potentially enhance user engagement.

Strength:
1. Providing application in
a strategy for reinforcing tweets to

increase the engagement rate is interesting.
2.  The paper investigates the Natural Language Processing (NLP) has been instrumental in deriving insights from posts, various fact-checking entities on Twitter, such as @snopes, @PolitiFact, and @APFactCheck, adopt diverse strategies.

3. The paper includes Descriptive Statistics for Fact-Check Spread Profile on Twitter.

Weakness:
1. The abstract needs to rewrite and the statement "
This study aims to underline the significance of these nuances to ensure that authentic information not only reaches its audience but also deeply resonates with them." at the need of the abstract must be move to the objective paragraph in page 3 before the Literature Survey.

2. The novelty of this research paper is weak as the authors only focus on fundamental analysis.

Author Response

Thank you for your insightful feedback on our manuscript. We have sought to follow your considerations carefully. The responses are below, and the effects in the paper are highlighted by the blue notes.

Weakness:
1a. The abstract needs to rewrite and

The abstract is re-written.

1b. the statement "This study aims to underline the significance of these nuances to ensure that authentic information not only reaches its audience but also deeply resonates with them." at the need of the abstract must be move to the objective paragraph in page 3 before the Literature Survey.

Done.

2. The novelty of this research paper is weak as the authors only focus on fundamental analysis.

Thanks for your perspective on the fundamental nature of the provided results and agreed that the paper builds upon established frameworks and theories. However, we would like to highlight its novel aspects. Firstly, our paper synthesizes a broad range of recent studies to provide a comprehensive understanding of the role of emotional drivers in the quite unexplored filed of fact-checking spreding. This synthesis, we believe, offers a unique contribution by integrating diverse findings into a coherent framework, which has not been extensively explored in previous research. Secondly, while our approach may seem fundamental, we have applied these principles in innovative ways to address contemporary challenges in fact-checking spread. The proposed approach incorporates the latest advancements in text reinforcement techniques, tailored specifically for mitigating the spread of fake news in the digital age. This application of fundamental theories to modern-day problems, adds a fresh perspective to the existing body of knowledge. Lastly, the practical implications of our research are significant. By focusing on the cognitive underpinnings of misinformation spread, the paper opens new avenues for developing more effective strategies for fact-checking spread, which are crucial in today's misinformation landscape. We hope that these points address your concerns regarding the novelty of our paper. We are open to further suggestions and are committed to enhancing the value of our work based on your valuable feedback.

Reviewer 3 Report

Comments and Suggestions for Authors

The manuscript presents a comprehensive study on the optimization of fact-check dissemination by integrating emotional and mental nuances in technological approaches. The paper is well-organized and addresses an important topic in the realm of information dissemination on social media platforms. The use of cognitive reinforcement techniques to potentially enhance user engagement is a novel approach and is well articulated in the paper.

The literature review is thorough, providing a clear background on the current state of fact-checking dissemination and highlighting the importance of tone and emotional appeal in engaging users. The methodology section is detailed, and the cognitive reinforcement strategies are well explained, offering insights into how factual information can be made more engaging to users.

However, the paper could be strengthened by including a more robust comparison with existing baseline models. While the proposed model shows promise, it is essential for readers to understand how it stands against other established models in the field. The current lack of comparison makes it difficult to evaluate the full potential and efficacy of the suggested approach.

Additionally, the results and discussion sections are informative, but they would benefit from a deeper analysis of the statistical significance of the findings. This could provide a clearer picture of the practical implications of the research.

Minor revisions are suggested to address these points:

1. Comparative Analysis: Expand the discussion on the comparative analysis of the proposed model against existing baseline models. This should include a broader range of models and a clear presentation of metrics that demonstrate the proposed model's performance relative to the baselines.

2. Statistical Analysis: Provide additional statistical analysis to support the findings, especially concerning the engagement rates and the effectiveness of the cognitive reinforcement strategies.

3. Broader Implications: Discuss the broader implications of the findings for other social media platforms and in various contexts, such as different cultures or types of misinformation.

4. Limitations and Future Work: A more detailed discussion on the limitations of the current study and potential areas for future research would be valuable. This could include how the model might adapt to evolving social media landscapes and misinformation tactics.

In summary, this paper is a fair contribution to the field, and with these minor revisions, it could provide a more compelling case for the use of cognitive reinforcement in fact-check dissemination.

Comments on the Quality of English Language

English expression is in general okay in this paper.

Author Response

Thank you for your insightful feedback on our manuscript. We have sought to follow your considerations carefully. The responses are below, and the effects in the paper are highlighted by the pink notes.

Minor revisions are suggested to address these points:

1. Comparative Analysis: Expand the discussion on the comparative analysis of the proposed model against existing baseline models. This should include a broader range of models and a clear presentation of metrics that demonstrate the proposed model's performance relative to the baselines.

We acknowledge your suggestion for a more robust comparison with existing baseline models. However, as indicated in our literature review, our research enters relatively uncharted territory regarding the application of cognitive reinforcement techniques in fact-check dissemination. This unique focus has limited the availability of directly comparable baseline models. Our approach is one of the first of its kind, integrating emotional and mental nuances in technological approaches for fact-check dissemination. While we understand the importance of comparative analysis, the innovative nature of our model poses a challenge in finding established models with sufficiently similar methodologies and objectives for a direct comparison. We believe this highlights the novel aspect of our research and opens avenues for future studies to establish baselines against which our model can be evaluated.

2. Statistical Analysis: Provide additional statistical analysis to support the findings, especially concerning the engagement rates and the effectiveness of the cognitive reinforcement strategies.

Additional information has been included.

3. Broader Implications: Discuss the broader implications of the findings for other social media platforms and in various contexts, such as different cultures or types of misinformation.

An additional section and some paragraphs are included throughout the text to discuss such implications.

4. Limitations and Future Work: A more detailed discussion on the limitations of the current study and potential areas for future research would be valuable. This could include how the model might adapt to evolving social media landscapes and misinformation tactics.

Additional information has been included.

Reviewer 4 Report

Comments and Suggestions for Authors

Provide more details about the methodology used for analyzing Twitter data. Consider adding a comparative analysis with other social media platforms to understand if the findings are unique to Twitter or applicable more broadly. Include specific case studies or examples where emotional nuances in fact-checking have been effective. This can provide practical insights and strengthen the argument. Utilize more graphs and visual aids to represent data. This can make the paper more engaging and easier to understand for readers not specialized in the field.

Author Response

Thank you for your insightful feedback on our manuscript. We have sought to follow your considerations carefully. The responses are below, and the effects in the paper are highlighted by the yellow notes.

1. Provide more details about the methodology used for analyzing Twitter data.

Additional information has been included.

2. Consider adding a comparative analysis with other social media platforms to understand if the findings are unique to Twitter or applicable more broadly.

The idea of expanding the scope to understand whether our findings are unique to Twitter or applicable more broadly is indeed insightful and would undoubtedly add value to the research in this field but as a future work (this remark is included in the conclusion). After careful consideration, the authors concluded that the research design and methodology were specifically tailored to the unique environment and user dynamics of Twitter, which presents distinct characteristics relevant to the study's focus. The platform's structural and functional aspects, and mechanisms, are relevant to our analysis and may not directly translate to other platforms. In addition, conducting a robust comparative analysis across multiple platforms would require a significant expansion of the research scope, including additional data collection, analysis, and potentially different methodological approaches. This would go beyond the current aims and constraints of this paper, including time and resource limitations. Moreover, the authors believe that focusing in-depth on a single platform allows for a more nuanced and detailed understanding of the specific phenomena under study. It enables us to provide richer insights and more specific recommendations tailored to the Twitter platform, which could serve as a foundational basis for future research that could undertake the comparative analysis you suggest.

3. Include specific case studies or examples where emotional nuances in fact-checking have been effective. This can provide practical insights and strengthen the argument.

It is not easy to find research papers as such focused on fact-checking, thus the requested information are provided but from the fake-news perspective.

4. Utilize more graphs and visual aids to represent data. This can make the paper more engaging and easier to understand for readers not specialized in the field.

Additional plots has been included.